

# The complex role of transcription factor GAGA in germline death during *Drosophila* spermatogenesis: transcriptomic and bioinformatic analyses

Svetlana Fedorova[1], Natalya V. Dorogova[1], Dmitriy A. Karagodin[1], Dmitry Yu Oshchepkov[2], Ilya I. Brusentsov[1], Natalya V. Klimova[3] and Elina M. Baricheva[1]

[1] Department of Cell Biology, Institute of Cytology and Genetics SB RAS, Novosibirsk, Russian Federation
[2] Department of Systems Biology, Institute of Cytology and Genetics SB RAS, Novosibirsk, Russian Federation
[3] Department of Molecular Genetics, Institute of Cytology and Genetics SB RAS, Novosibirsk, Russian Federation

Corresponding author
Svetlana Fedorova,
fsveta@bionet.nsc.ru

## ABSTRACT

The GAGA protein (also known as GAF) is a transcription factor encoded by the *Trl* gene in *D. melanogaster*. GAGA is involved in the regulation of transcription of many genes at all stages of fly development and life. Recently, we investigated the participation of GAGA in spermatogenesis and discovered that *Trl* mutants experience massive degradation of germline cells in the testes. *Trl* underexpression induces autophagic death of spermatocytes, thereby leading to reduced testis size. Here, we aimed to determine the role of the transcription factor GAGA in the regulation of ectopic germline cell death. We investigated how *Trl* underexpression affects gene expression in the testes. We identified 15,993 genes in three biological replicates of our RNA-seq analysis and compared transcript levels between hypomorphic $Trl^{R85}/Trl^{362}$ and *Oregon* testes. A total of 2,437 differentially expressed genes were found, including 1,686 upregulated and 751 downregulated genes. At the transcriptional level, we detected the development of cellular stress in the *Trl*-mutant testes: downregulation of the genes normally expressed in the testes (indicating slowed or abrogated spermatocyte differentiation) and increased expression of metabolic and proteolysis-related genes, including stress response long noncoding RNAs. Nonetheless, in the Flybase Gene Ontology lists of genes related to cell death, autophagy, or stress, there was no enrichment with GAGA-binding sites. Furthermore, we did not identify any specific GAGA-dependent cell death pathway that could regulate spermatocyte death. Thus, our data suggest that GAGA deficiency in male germline cells leads to an imbalance of metabolic processes, impaired mitochondrial function, and cell death due to cellular stress.

## INTRODUCTION

Cell death occurs in any living organism. Cell death can be accidental, *i.e.,* caused by physical, chemical, toxic, or mechanical damage, or regulated, caused by intracellular problems or exposure to some factors in the extracellular microenvironment. The cell first attempts to cope with such perturbations and to restore cellular homeostasis, but if they are too strong or prolonged and cannot be reversed, then the process of elimination of the potentially dangerous cell is triggered and it dies. Regulated cell death can also be involved in an organism's development or tissue renewal program; this completely physiological type of death is called programmed cell death (*Galluzzi et al., 2018*). In 2018, the Nomenclature Committee on Cell Death (NCCD) proposed a classification of 12 major cell death subroutines, based on molecular, morphological, and biochemical characteristics (*Galluzzi et al., 2018*). The best-known, "classic" types of death are apoptosis, necrosis, and autophagic death. A key feature of apoptotic cell death is the activation of caspases, apoptosome assembly, mitochondrial outer-membrane permeabilization, DNA condensation and specific fragmentation, membrane blebbing, and apoptotic-body formation (*Yacobi-Sharon, Namdar & Arama, 2013*; *Galluzzi et al., 2018*). Necrosis is characterized by membrane damage, a release of calcium ions into the cytoplasm, bloating of organelles, and acidification of the cytoplasm. Necrosis is often accompanied by inflammation. Autophagic cell death is defined by the accumulation of autophagosomes: double-membrane vesicles that implement cell self-digestion by sequestering cellular materials to lysosomes (*Yacobi-Sharon, Namdar & Arama, 2013*).

In *Drosophila* testes, ∼20–30% of spermatogonial cysts undergo spontaneous cell death: germline cell death. It takes place in the apical part of a testis before meiosis and has morphological features of both apoptosis and necrosis but not autophagic cell death (*Yacobi-Sharon, Namdar & Arama, 2013*). Germline cell death is regulated by mitochondrial serine protease HtrA2/Omi but is independent of effector caspases. Bcl-2 family proteins Debcl and Buffy and mitochondrial nuclease EndoG are associated with germline cell death (*Yacobi-Sharon, Namdar & Arama, 2013*). It was also shown that defective spermatocytes can be eliminated by p53-mediated programmed necrosis (*Napoletano et al., 2017*).

Previously, we have investigated the causes of fertility loss in *Trl*-mutant males and found that spermatocytes undergo mass death (*Dorogova et al., 2014*; *Dorogova et al., 2021*). The *Trl* gene encodes transcription factor GAGA (also known as GAF), which participates in the regulation of the transcription of a large group of genes with different cellular functions in *D. melanogaster* (*Van Steensel, Delrow & Bussemaker, 2003*; *Omelina et al., 2011*). Various studies have revealed that GAGA is required for embryogenesis and eye and wing development in *Drosophila* (*Farkas et al., 1994*; *Bhat et al., 1996*; *Dos-Santos et al., 2008*; *Omelina et al., 2011*; *Bayarmagnai et al., 2012*; *Fedorova et al., 2019*). We have researched the role of GAGA in the development and function of the *Drosophila* reproductive system and found that GAGA is required for gonadogenesis (*Dorogova et al., 2014*; *Fedorova et al., 2019*). We have demonstrated that *Trl* underexpression causes multiple disorders of oogenesis and spermatogenesis, resulting in a significant loss of

fertility (*Dorogova et al., 2014*; *Fedorova et al., 2019*). Additionally, we have reported that mass autophagic death of germline cells occurs in *Trl*-mutant testes during spermatogenesis (*Dorogova et al., 2014*; *Dorogova et al., 2021*). The question has arisen as to what causes the death of spermatocytes. Because the GAGA protein is a transcription factor, can it regulate the activity of cell death genes? Or does death result from dysregulation of transcription of multiple genes and from metabolic disorders? In this study, we tried to elucidate the mechanisms triggering mass death in the testes of hypomorphic *Trl* mutants. For this purpose, we performed transcriptomic profiling of *Trl*-mutant and normal (control) *D. melanogaster* testes.

Accordingly, we report results of high-throughput RNA sequencing (RNA-seq). We found 4,300 differentially expressed genes (DEGs) in hypomorphic *Trl* mutants. Nonetheless, we failed to identify a specific signaling cascade whose activation could lead to the germline cell death in *Trl* mutants. Therefore, we believe that the mass degradation of *Trl* spermatocytes represents regulated cell death caused by the cellular stress that is a consequence of imbalanced intracellular processes, primarily metabolism and mitochondrial activities.

## MATERIALS & METHODS

### Flies

The mutant null-allele $yw;Trl^{R85}/Sb$ $Ser$ $y+$ strain was kindly provided by Dr. F. Karch (University of Geneva, Switzerland) and described previously (*Farkas et al., 1994*). $Trl^{R85}/Trl^{R85}$ mutants are not viable in the homozygous state and do not survive to the imago stage; therefore, in our work, we used a combination of the $Trl^{R85}$ allele with a strong hypomorphic mutation ($Trl^{362}$) that disrupts the 5′ region of the gene (*Ogienko et al., 2006*). *Oregon-R-modEncode* (cat. #25211, Bloomington, USA) served as the wild-type strain. All *Drosophila* stocks were raised at 25 °C on a standard cornmeal medium.

### RNA-seq

For RNA-seq experiments, three biological replicates were performed. For each RNA-seq assay, total RNA was isolated from testes of 100 1–2-day-old male control flies (*Oregon R*) and from the same number of $yw; Trl^{R85}/Trl^{362}$ testes. Young males (1–2 days old) were anesthetized and placed into Hanks' solution (BioloT, Russia). Testes were removed from the flies using dissecting needles, teased apart from seminal vesicles, and immediately transferred into a microcentrifuge tube placed in liquid nitrogen. Fifty testes were put in each tube and stored in liquid nitrogen. Total RNA was isolated using the TRIzol reagent (Invitrogen, Waltham, MA, USA). After that, genomic DNA was extracted from the organic phase and was tested by PCR for the absence of normal copies of the *Trl* gene in mutant samples. PCR was carried out in a 20 μl reaction mixture consisting of $1 \times$ PCR-buffer [16 mM $(NH_4)_2SO_4$; 67 mM Tris–HCl pH 8.9 at 25 °C; 0.1% of Tween 20], 1.5 mM $MgCl_2$, 0.2 mM each dNTPs, 1 U of Taq DNA Polymerase (Biosan, Novosibirsk, Russia), and 0.5 μM each primer. The following primers were employed: ex1a (5′-agttatccaacgttggcgag-3′) and NC70-1 (5′-cagacgttagttattagctc-3′). The PCR products were separated by electrophoresis on a 1% agarose gel in $0.5 \times$ TBE buffer with ethidium bromide staining.

**Table 1  RNA-seq data alignment statistics.**

| Library | Number of reads | Number of uniquely mapped reads | Number of reads mapped to multiple loci | Number of reads mapped to too many loci |
|---|---|---|---|---|
| 362-testes1.tr | 41,397,803 | 39,832,848 (96.22%) | 1,209,655 (2.92%) | 86,802 (0.20%) |
| 362-testes2.tr | 42,823,074 | 40,781,860 (95.23%) | 1,627,713 (3.80%) | 117,174 (0.25%) |
| 362-testes3.tr | 36,436,374 | 34,488,834 (94.65%) | 1,564,303 (4.29%) | 104,373 (0.27%) |
| Or-testes10.tr | 37,418,763 | 36,165,776 (96.65%) | 971,386 (2.60%) | 94,938 (0.23%) |
| Or-testes11.tr | 37,927,024 | 36,548,954 (96.37%) | 1,120,371 (2.95%) | 144,254 (0.34%) |
| Or-testes12.tr | 38,382,924 | 36,977,384 (96.34%) | 1,110,115 (2.89%) | 144,196 (0.40%) |

The quality of the total-RNA samples was evaluated using a Bioanalyzer 2100 device (Agilent, Santa Clara, CA, USA). Samples with optimal RNA integrity numbers (RINs) were chosen for further analysis. Additionally, the total RNA was quantitated on an Invitrogen Qubit$^{TM}$ 2.0 fluorometer (Invitrogen, Waltham, MA, USA). Total RNA (1.2 µg) was treated with DNase (QIAGEN RNase-Free DNase Set, Hilden, Germany) and purified on PureLink$^{TM}$ RNA Micro Kit columns (Invitrogen, Waltham, MA, USA). RNA-seq libraries were prepared from 0.3 µg of total RNA by means of the TruSeq® Stranded mRNA LT Sample Prep Kit (Illumina, San Diego, CA, USA) according to the manufacturer's instructions for barcoded libraries. The quality of the obtained libraries was checked on Bioanalyzer 2100 and with the DNA 1000 Kit (Agilent, Santa Clara, CA, USA). After normalization, barcoded libraries were pooled and sequenced on a NextSeq550 instrument using NextSeq® 550 High Output v2.5 Kit 75 Cycles (Illumina, San Diego, CA, USA).

The quality of the obtained raw Fastq files was tested and analyzed in FastQC. To improve the quality of the raw reads, we employed the TrimGalore software, v.0.6.7 (https://www.bioinformatics.babraham.ac.uk/projects/trim_galore/), Cutadapt v.1.15, and FastQC v.0.11.5 *via* these procedures: removal of a base from either the first or end position if the quality was low (Phred score: 20), trimming of Illumina adapters, and removal of any remaining reads that are <20 bases long. The trimmed reads were aligned with the annotated *D. melanogaster* genome retrieved from FlyBase (http://flybase.org/, dmel_r6.34). The alignment was performed in STAR aligner v.2.7.5a (*Dobin et al., 2013*) with determination of the number of reads per gene (options -quantMode GeneCounts) (Table 1).

Differential expression analysis was performed in DESeq2 (*Anders & Huber, 2010*) on the IRIS web server, which is publicly available at (*Monier et al., 2019*). Genes were considered differentially expressed if their average change of expression was greater than 2-fold and a Benjamini–Hochberg-adjusted *P*-value ($P_{adj}$) was less than 0.05 to ensure statistical significance (*Anders & Huber, 2010*). Principal component analysis of normalized log-transformed read counts was performed by means of DESeq2 (*Anders & Huber, 2010*) on the IRIS web server to determine the reproducibility of the analyzed replicates (Fig. S1).

We used the scottyEstimate function of Scotty (*Busby et al., 2013*) and ssizeRNA 1.3.2 (*Bi & Liu, 2016*) to measure the statistical power of the differential expression study. We used the Scotty program with the following parameters: fc =2, pCut = 0.05,

minPercDetected = 80, maxReps = 10, minReadsPerRep = 10,000,000, maxReadsPerRep = 100,000,000, minPercUnbiasedGenes = 50, pwrBiasCutoff = 50, and alignmentRate = 98. Parameters for the ssizeRNA program were as follows: nGenes = 15,000, pi0 = 0.83, $m$ = 200, mu = mu1, disp = disp1, fc = 2, fdr = 0.05, power = 0.8, maxN = 15. Both approaches showed a calculation power of more than 0.8 (Fig. S2), which means that the 3 biological repeats we used for experiment and control group are sufficiently statistically powerful to detect at least 80% of differentially expressed genes.

The RNA-seq data have been deposited in the NCBI repository and can be accessed with the BioProject accession ID PRJNA785453.

## Validation of RNA-seq data by quantitative PCR (qPCR)

For qPCR validation, three biological replicates for controls and mutants with two technical replicates per experiment were analyzed. For qPCR, RNA was isolated as described in the transcriptome RNA-seq assay. For each biological replicate, total RNA was isolated from 100 testes of 1–2-day-old *Oregon R* males or from the same number of *yw; Trl$^{R85}$/Trl$^{362}$* testes.

Total RNA was isolated with the TRIzol reagent (cat. #15596026; TRIzol$^{TM}$ Reagent, Invitrogen, Waltham, MA, USA) and cleared by means of magnetic beads (#A63987; RNAClean XP, Beckman, Brea, CA, USA) according to the manufacturer's instructions. After that, RNA was incubated with DNase I (#MAN0012000; Thermo Scientific, Waltham, MA, USA) and precipitated with ethanol. cDNA synthesis was performed with Maxima H Minus Reverse Transcriptase (#EP0751; Thermo Scientific, Waltham, MA, USA) and oligo-dT$_{20}$. qPCR was carried out on a CFX96 Real-Time System (BioRad, Hercules, CA, USA). The thermal cycling protocol for the amplification reaction was as follows: 5 min preincubation at 95 °C; next, 40 cycles of 10 s at 95 °C and 30 s at 60 °C; followed by a melting curve program at 65–95 °C. Primers used in the qPCRs are listed in Table S1, and the verification data are represented in Fig. S3 and Table S2.

## Bioinformatics

The search for GAGA-binding sites was performed with the help of the SITECON software package (*Oshchepkov et al., 2004*) *via* a previously developed computational approach (*Omelina et al., 2011*).

Gene Ontology (GO) enrichment analysis was conducted on the FlyEnrichr server (*Chen et al., 2013*; *Kuleshov et al., 2016*) and on the GOrilla server (*Eden et al., 2007*; *Eden et al., 2009*). For GOrilla, the *P*-value threshold was set to $10^{-3}$. The sets of up- and downregulated genes ($P_{adj} \leq 0.05$, fold change [FC] >2 or <0.5) were analyzed for GO enrichment in relation to the genes expressed in testes (14,579 genes).

## RESULTS

We performed RNA-seq of poly(A)+ RNA from the testes of hypomorphic *Trl$^{R85}$/Trl$^{362}$* mutant males. The testes of the wild-type *Oregon* line served as a control. In three biological replicates, we identified 15,993 genes out of the 17,612 genes annotated in FlyBase (http://flybase.org/, dmel_r6.34). After testing for differential expression and exclusion

of weakly expressed genes (total read count <10), 14,579 genes were chosen for further analysis. *Vedelek et al. (2018)* published a *Drosophila* testis transcriptome containing 15,015 genes. *Vedelek et al. (2018)* cut the testes into three parts (apical, middle, and basal regions) and identified transcription profiles for genes in each testis region. Our RNA-seq results almost completely matched the testis RNA-seq data from ref. *Vedelek et al. (2018)*, proving the reliability of our RNA-seq results.

Then, we compared transcript levels between $Trl^{R85}/Trl^{362}$ and *Oregon R* testes. RNA-seq analysis revealed 2,437 differentially expressed genes (DEGs) ($P_{adj} \leq 0.05$ and FC $\geq$ 2) in the testes of *Trl* mutants, where 1,686 transcripts were upregulated and 751 were downregulated.

## Gene ontology analysis of RNA-Seq data

We used online tools FlyEnrichr and GOrilla to search for GO terms enriched in our dataset (*Eden et al., 2007*; *Eden et al., 2009*; *Chen et al., 2013*; *Kuleshov et al., 2016*). For the GO analysis, we subdivided the testis DEGs into upregulated and downregulated. We conducted the analysis in all three GO categories (biological processes, molecular functions, and cellular components) and compared the corresponding datasets to each other. The two tools, FlyEnrichr and GOrilla, yielded similar results (Figs. S4 and S5). As expected, analysis by biological processes showed that the set of upregulated genes was especially enriched with genes related to reproduction and development (Figs. S4 and S5). In addition, in the set of upregulated genes, we detected enrichment with metabolic and proteolysis-related genes. The 43 genes involved in proteolysis (of which four are testis-specific genes) include genes encoding enzymes with serine-type peptidase activity (17 genes), 11 other endopeptidase genes, and 15 protease genes associated mainly with the catabolism of ubiquitinated proteins.

The downregulated DEG set showed the expected enrichment with proliferation and differentiation genes, consistently with the reduced size of the testes in the *Trl* mutants. These include seven dynein complex–related genes (*Dhc36C*, *Dhc98D*, *Dhc62B*, *Dhc64C*, *Dnah3*, *sw*, and *Sdic1*) whose expression diminished 2–6-fold. Dyneins are ATPases and explain the enrichment with the GO term "ATP-dependent microtubule motor activity, minus-end-directed". Four (*Dhc36C*, *Dhc62B*, *Dnah3*, and *Sdic1*) of these seven genes are testis-specific, and *sw* is involved in sperm individualization.

Unexpectedly, enrichment with the GO term "lysosomes" was detected among both up- and downregulated DEGs. Genes encoding enzymes of catabolic processes, primarily endopeptidases, predominated among the upregulated DEGs associated with the GO term "lysosome". The second place—among the DEGs that decreased their expression in the *Trl* mutant testes—belongs to the GO term "mannose metabolic process", represented by lysosomal mannosidases LManIII, LManIII, LManIV, LManV, and LManVI (whose expression declined 2.0-38.5-fold) and alpha-Man-Ic (downregulated by 3.2-fold), which is presumably associated with the endoplasmic reticulum and Golgi membrane. There are no testis-specific genes among these genes, but most of them are associated with lysosomes.

## Tissue specificity of the DEGs

To identify the tissue specificity of gene expression, we used data from *Li et al. (2014)* and *Vedelek et al. (2018)*, who have determined the specificity of expression by means of the modENCODE tissue expression database. In their work, tissue specificity scores higher than 4 represent testis-enriched genes and lower values represent genes expressed in multiple tissues or ubiquitously; negative values denote under-representation of transcripts in testes (*Vedelek et al., 2018*). They identified 2,602 testis-enriched genes in the *Drosophila* testis transcriptome, including 682 long noncoding RNAs (lncRNAs) (*Vedelek et al., 2018*). The present RNA-Seq data contain genes with different tissue specificity: 3,964 genes expressed in several or many tissues, 2,454 testis-enriched genes, and 7,330 genes under-represented (*i.e.,* normally not expressed) in testes (Fig. 1). We compared our transcriptome results with *Vedelek et al.'s (2018)* RNA-seq results and noticed a 94.31% overlap between the testis-enriched gene sets (2,454 testis-enriched genes including 625 lncRNAs). Among our DEGs, we found 1,565 genes expressed in two or more tissues; 681 of them were upregulated and 884 were downregulated. As many as 997 testis-enriched genes, including 303 lncRNAs, featured differential expression in mutant testes: 152 of testis-enriched genes were upregulated (69 twofold or more) and 845 were downregulated (235 genes: twofold or more) (Table S3). It should be noted that among the genes usually under-represented in testes (*i.e.*, genes with the negative value of the tissue specificity index (*Li et al., 2014*; *Vedelek et al., 2018*)), the expression pattern was exactly the opposite. In RNA-seq data from *Trl* mutants, 8,721 genes were found to be under-represented in the testes, of which 2,636 were DEGs; 1,836 of them were upregulated, whereas 800 were downregulated (Fig. 1). It is worth pointing out that the expression of late spermatogenesis genes necessary for sperm formation and maturation (Fig. 2A) was the lowest in the *Trl* mutant testes, while the expression of metabolic genes was higher than normal (Fig. 2B).

## lncRNAs

Among the 2,438 testis DEGs, 27.7% (675) were lncRNAs, of which 340 lncRNAs (20.17% of all overexpressed transcripts) showed increased expression, and 219 (29.2% of all downregulated transcripts) proved to be downregulated. Because mutations in the *Trl* gene affect germline cells and lead to reduced fertility and spermatogenesis abnormalities (*Dorogova et al., 2014*), we were primarily interested in expression alterations of testis-specific lncRNAs. In our data, we identified 625 lncRNA genes that, as reported by *Li et al. (2014)* and *Vedelek et al. (2018)*, can be assigned to testis-enriched genes, 217 of which were differentially expressed. The functions of most of testis-enriched lncRNAs are not known, but *Wen et al. (2016)* screened 128 testis-specific lncRNAs and found 33 lncRNAs that perform critical functions in the regulation of late spermatogenesis and whose knockouts cause spermatogenesis defects and consequent loss of male fertility. According to our RNA-seq dataset, for three (CR43633, CR43634, and CR43414) of those 33 testis-specific lncRNAs, expression increased by 6-fold or more, whereas for eight (CR43753, CR44412, CR44344, CR43839, CR43862, CR44371, TS15, and CR43416) expression diminished by ≥2-fold. Thus, the knockdown of the above eight RNAs in the testes of *Trl* mutants may be responsible for at least some of the spermatogenesis abnormalities observed.

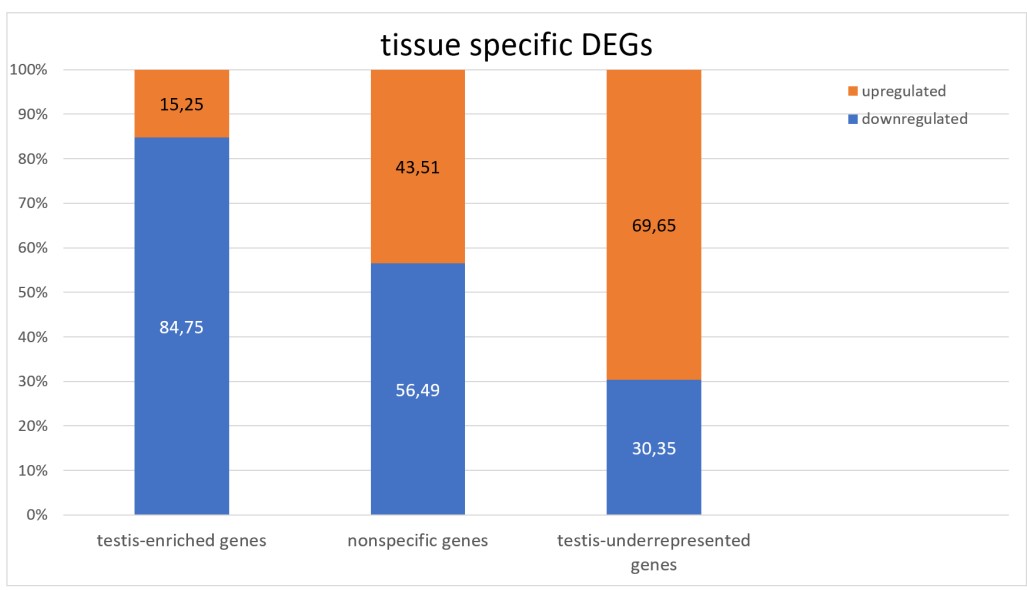

**Figure 1** **Comparison of the expression of testis-specific and nonspecific genes.**

In addition, in mutant organs, we identified 76 lncRNAs with significantly altered expression that are not expressed in normal testes (*Wen et al., 2016*; *Vedelek et al., 2018*). It should be noted that CR34262, CR32010, and CR45346 were among them and featured an increase in expression by 104-, 52-, and 51-fold, respectively; it is known about these three that they are upregulated in response to environmental stressors (*Brown et al., 2014*; *Wen et al., 2016*). CR44138—which is induced most strongly by paraquat treatment (oxidative stress) followed by caffeine, Cd, Cu, and Zn—as well as stress-response related CR45346 (*Brown et al., 2014*; *Wen et al., 2016*) were also upregulated in *Trl*-mutant testes, by 45-fold and 25-fold, respectively.

## Mitochondrial function and structure

Earlier, we showed that *Trl*-mutant spermatocytes acquire abnormal mitochondrion structure and morphology before dying (*Dorogova et al., 2014*). Here, we examined the connection of our transcriptome dataset with mitochondria (Table S3). In our dataset, we found 778 genes associated with the FlyBase GO term ''mitochondrion'' (GO:0005739); for 267 of them, expression changed significantly: 123 manifested higher expression, with 35 being upregulated twofold or more, and 144 were downregulated, with 13 downregulated ≥2-fold (Table S3).

Given that our previously examined mutant mitochondrial phenotype suggested that mitochondrial permeability and function were impaired, we next analyzed the genes encoding respiratory chain components, *i.e.,* the FlyBase GO ''respiratory chain complex'' term (GO:0098803). Seventy-seven genes out of 79 turned out to be expressed in testes, of which 16 genes were significantly upregulated (five, including Cyt-c1, twofold or more), and eight genes were downregulated. It is worth noting that although for most of the respiratory-chain genes, there was no significant change in expression, some of the

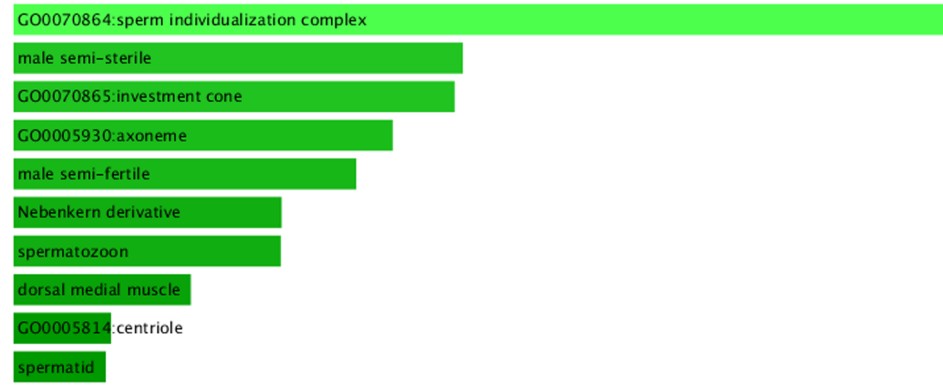

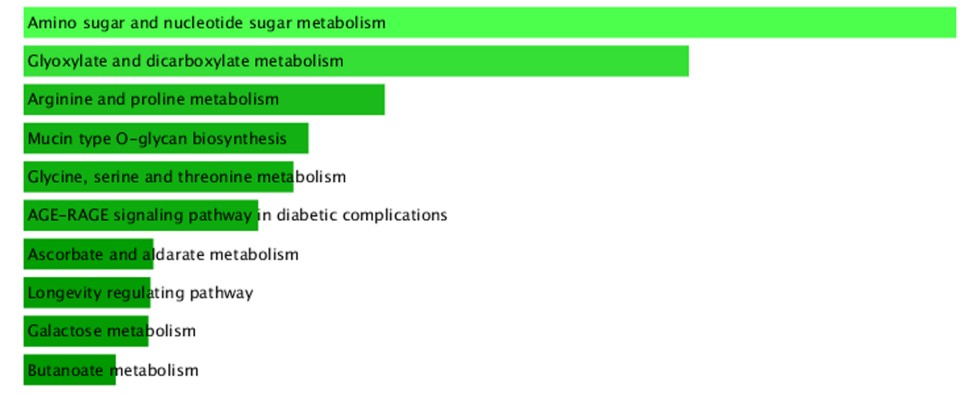

**Figure 2** **FlyEnrichr GO analysis of testis-specific and nonspecific genes.** (A) Top GO terms enriched among testis-enriched downregulated genes. FlyEnrichr GO analysis, sorted by combined score. (B) GO terms most enriched among upregulated genes under-represented in testes. FlyEnrichr GO analysis, sorted by combined score.

subunits underwent expression changes in opposite directions in every component of the respiratory complex and together may have led to an imbalance in the functioning of the respiratory chain as a whole.

## Cell death genes

Lack of GAGA in testes leads to death and degradation of germline cells (*Dorogova et al., 2021*). The question we wanted to answer in this project is whether GAGA is a transcriptional regulator of cell death in *Drosophila* testes. To this end, we compared the group of FlyBase GO lists of death-related genes (corresponding to terms "cell death" GO:0008219, "apoptotic process" GO:0006915, "programmed cell death" GO:0012501, and "autophagic cell death" GO:0048102) with the RNA-seq data we obtained (Table S3, Fig. 3).

Overall, less than half cell death genes underwent changes in expression in the *Trl*-mutant testes. There were twice as many downregulated genes as upregulated genes

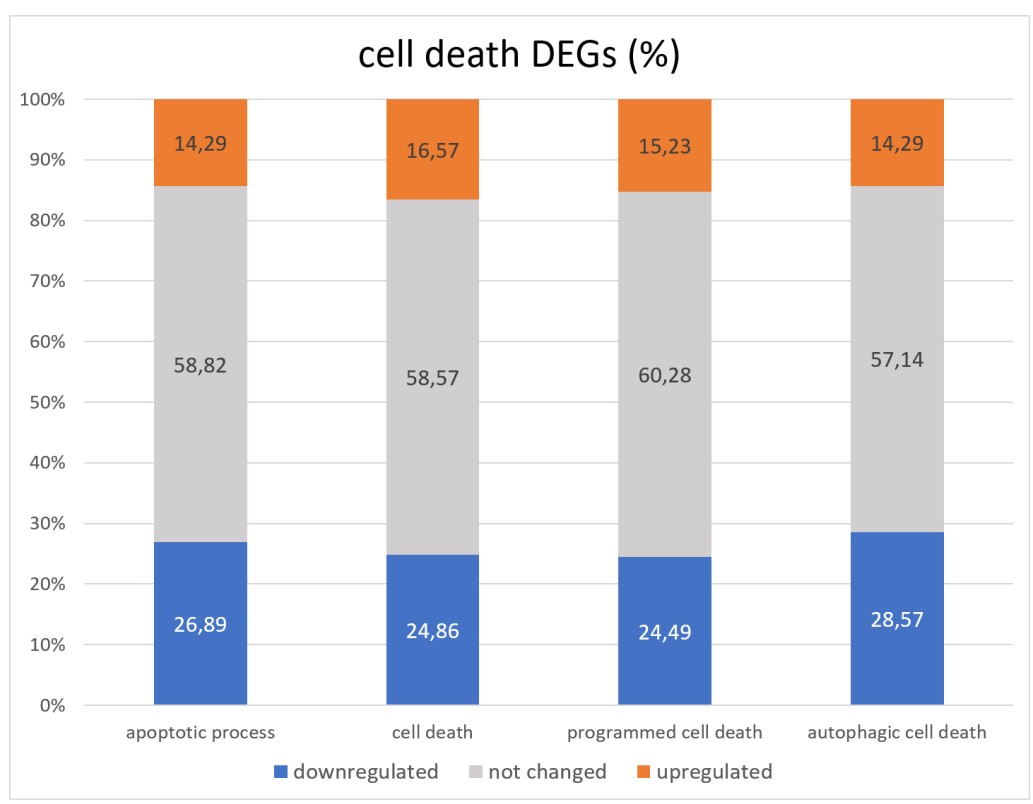

**Figure 3** Distribution of genes associated with GO cell death related terms with increased, decreased, or unchanged expression levels.

(Fig. 3). The "cell death" GO dataset consists of 399 genes, and 362 were found in our RNA-seq data; only 150 of them were differentially expressed in *Trl*-mutant testes, 10 genes were downregulated twofold or more: *ninaE, Dark, CG14118, Trf2, Cdk5alpha, HUWE1, CG9593, Mcm10, Dcr-2,* and *CG30428*; 22 genes proved to be overexpressed ≥2-fold: *rst, ix, pnt, NijA, Drep2, Fhos, wrapper, Eip74EF, DNaseII, Corp, Abd-B+H383A1H4:H14565, CG5860, CG2918, CG31928, Buffy, en, Mdh2, IFT57, ft, lncRNA:Hsromega, tau,* and *Orct*. The most upregulated (FC = 146) was aspartic-type endopeptidase CG31928, which is active in lysosomes. From the 276 genes officially associated with "apoptotic processes" (according to GO FlyBase), our RNA-seq contained 238; only 98 (36.7%) were differentially expressed in *Trl*-mutant testes, and 13 of them were upregulated twofold or more: *Orct, wrapper, ft, lncRNA:Hsromega, DNaseII, Corp, Buffy, IFT57, Abd-B, CG2918, en, Drep2,* and *pnt*; seven genes were downregulated ≥2-fold: *Cdk5alpha, Dark, Mcm10, Dcr-2, CG14118, CG30428,* and *CG9593*. It must be mentioned that *lncRNA:Hsromega* is a stress-inducible lncRNA that takes part in the metabolism regulation as well (according to FlyBase). RNA-seq data from *Trl*-mutant testes contained 28 of the 29 genes from the "autophagic cell death" GO dataset, and 12 genes were differentially expressed; *Dark* and *Trf2* were downregulated and *Eip74EF* and *Mdh2* were upregulated twofold or more.

In our previous work, we demonstrated that autophagosomes and lysosomes are abundant in the cytoplasm of dying spermatocytes. Nevertheless, the death of these cells does not involve apoptosis (*Dorogova et al., 2021*). Here, we analyzed the genes belonging to the "autophagy" (GO:0006914) and "lysosome" (GO:0005764) FlyBase terms (Table S3). Only 71 out of 200 autophagy genes proved to be differentially expressed in *Trl*-mutant testes, seven of them were upregulated and three downregulated ≥2-fold. The most upregulated gene, *stj* (31-fold change) encodes a voltage-gated calcium channel subunit involved in the regulation of lysosomal fusion with endosomes and autophagosomes (*Tian et al., 2015*). The most interesting is the Bcl-2 family member, "Buffy", which participates in stress-induced cell death (*Sevrioukov et al., 2007*), including a lysosomal alternative germ cell death pathway in *Drosophila* (*Yacobi-Sharon, Namdar & Arama, 2013*). Among 106 lysosomal genes, approximately a half (46 genes) are differentially expressed in *Trl*-mutant testes, and three of the most overexpressed genes (CG31928: FC = 146, CG4847: FC = 13, CG5860: FC = 3.4) are endopeptidases involved in cell death.

Finally, we analyzed the dataset of the "cellular response to stress" (GO:0033554) term (Table S3). Approximately 30% of genes (240 out of 619, $P_{adj} < 0.05$) underwent changes in their expression; 30 genes were upregulated and 11 were downregulated in *Trl*-mutant testes. The most upregulated gene was *Hsp70Ba* (FC = 96); it encodes a protein involved in the response to heat shock and hypoxia and in the unfolded protein response (*Moutaoufik & Tanguay, 2021*). Besides, upregulated stress-responsive genes included cochaperone Hsc70-3 (FC = 2.5), the hypoxia-induced *CG2918* gene encoding a protein with unfolded-protein-binding activity, and the starvation-upregulated *Sirup* gene, which codes for a critical assembly factor for Complex II in the electron transport chain of mitochondria. It is noteworthy that the official "cellular response to stress" GO dataset do not include lncRNAs, whereas we mentioned above that at least five of the most upregulated (FC = 25 to 104) lncRNAs in our dataset are stress-inducible (*Brown et al., 2014*; *Wen et al., 2016*).

## GAGA target genes (analysis of GAGA-binding sites)

As mentioned above, our GO term analysis did not reveal enrichment of the DEG set with GO terms corresponding to cell death or stress. We decided to test 5′ regulatory regions of the genes (from FlyBase GO lists of genes) for the presence of GAGA-binding sites. For this purpose, we applied the SITECON software package (*Oshchepkov et al., 2004*) previously confirmed to recognize GAGA-binding sites (*Omelina et al., 2011*). Using SITECON software, we analyzed the presence of potential GAGA-binding sites of type 1 (GAGnGAG) and type 2 (GAGnnnGAG) in the −500...+1 region relative to the transcription start site (TSS). We also compared the obtained results with our findings about Flybase GO lists of genes related to development, spermatogenesis, and cellular stress (Fig. 4, Table S4) as well as all DEGs from *Trl*-mutant testes. Developmental genes tend to contain abundant GAGA-binding sites (*Van Steensel, Delrow & Bussemaker, 2003*; *Omelina et al., 2011*); therefore, we included the GO list of developmental genes in our analysis. As expected, the highest density of the GAGA-binding sites proved to be characteristic of genes of developmental processes, with GAGA-binding sites of the second type (GAGnnnGAG) occurring more

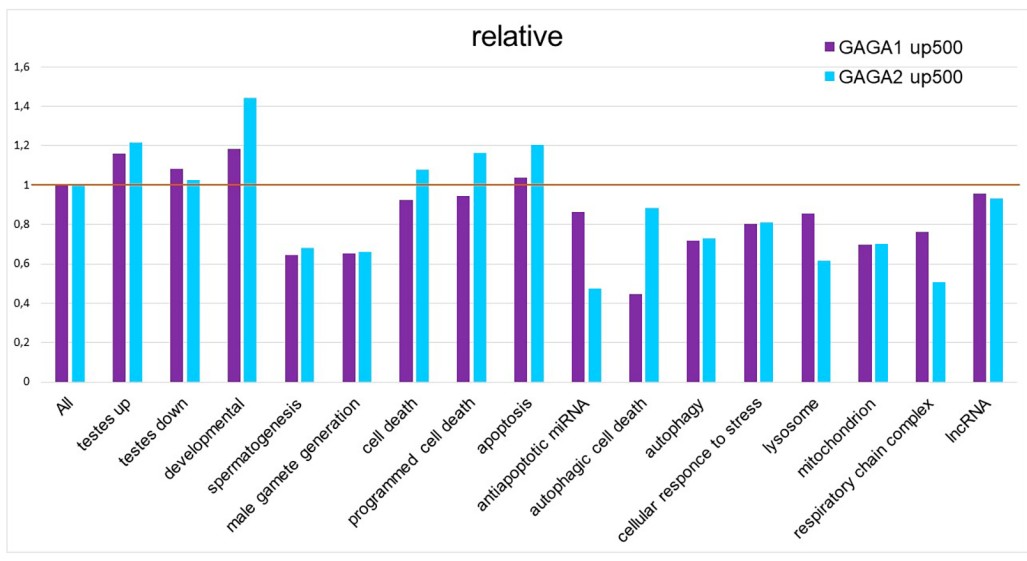

**Figure 4** **Relative density of probable GAGA-binding sites in various gene selections.** In the Flybase GO lists of genes, the figure presents the proportion of genes containing a GAGA-binding site(s) in the $-500\ldots+1$ region relative to the TSS, with normalization to the proportion of such genes among all *Drosophila* genes, according to SITECON results on type 1 GAGA-binding sites (<<GAGA1 up 500 >>) and type 2 GAGA-binding sites (<<GAGA2 up 500 >>).

frequently in these groups of genes (Fig. 4). In addition, the sets of DEGs identified by RNA-seq in our work also have higher density of GAGA-binding sites compared to the genome-wide average. In this work, we analyzed the testis transcriptome in hypomorphic *Trl* mutants, *i.e.,* having a lower amount of the GAGA transcription factor. Hence, DEG sets are expected to be enriched with genes regulated by this transcription factor, and the density of GAGA-binding sites in them should be higher than the genome-wide average. In 5′ (upstream) regions of the genes required for spermatogenesis or male gamete formation, the density of both types of GAGA-binding sites was found to be lower than the genome average.

A total of 27.7% of the DEGs in our transcriptome dataset were lncRNAs, we analyzed the lncRNA genes for potential GAGA-binding sites; the results are shown in Table S4 and Fig. 4. We found no enrichment with putative GAGA-binding sites in the set of the 675 ncRNAs expressed in hypomorphic *Trl* testes (Fig. 4). Out of the 217 testis-specific lncRNAs (*Vedelek et al., 2018*), only 13 genes had potential GAGA binding sites, out of the 383 testis-nonspecific lncRNA genes, 23 ones had putative GAGA binding sites. It is interesting to note that all of these GAGA-targeted testis-nonspecific lncRNAs increased their expression 3.7-fold or more in *Trl* mutants, *i.e.,* it can be assumed that their expression in the testes is normally suppressed by GAGA factor. Out of the 33 lncRNAs involved in the regulation of spermatogenesis (*Wen et al., 2016*), only four lncRNAs with potential GAGA-binding sites differ in expression between *Trl*-mutant and control testes: lncRNA:CR43414 is upregulated 6.36-fold in *Trl*-mutant testes, whereas lncRNA:CR43862, asRNA:TS15, and lncRNA:CR43753 are downregulated by 0.17-, 0.30-, and 0.37-fold, respectively.

Germline cells of $Trl^{R85}/Trl^{362}$ mutant testes die by autophagic death preserving the nuclear envelope intact, thus manifesting impaired mitochondrial morphology and an increased number of autolysosomes and lysosomes (*Dorogova et al., 2021*). We expected to find enrichment with GAGA-binding sites within 500 bp upstream regions of genes from these categories. Unexpectedly, the density of GAGA-binding sites in the official GO gene lists "autophagic cell death", "autophagy", "lysosome", "mitochondrion", and "respiratory chain complex" was lower than the genome-wide average, and genes related to cell death, programmed cell death, or apoptosis had slightly higher density of GAGA-binding sites (Fig. 4). Of note, type 2 GAGA-binding sites (GAGnnnGAG) are more frequent in cell death-related lists of genes in the GO database as well as among developmental genes in that database. We decided to clarify which genes from the cell death categories commonly contain GAGA-binding sites (Table S4).

The "apoptotic process" GO term corresponds to 267 genes in the Flybase, among which 63 genes (23.6%) contain type 1 GAGA-binding sites and 55 genes (20.6%) contain type 2 GAGA-binding sites (Table S4). Ninety genes out of 267 (33.71%) contain probable GAGA-binding sites in the −500...+1 region. Among these 90 genes, three functional groups of genes stood out: (1) 10 microRNA (miRNA, miR) genes (11.11%) that suppress apoptosis; (2) 11 genes (12. 22%) that regulate cell death in response to various stressors, and (3) seven genes related to DNA fragmentation, including five (*EndoG*, Testis EndoG-Like 3, *Drep1*, *Drep3*, and *CG14118*) encoding endonucleases and two coding for apoptotic executors (nbs and Corp) that recognize DNA breaks and inhibit apoptosis, thereby enabling repair. We identified the largest number of probable GAGA-binding sites in genes *Drep3* (14 sites), *scute* (13), *Egfr* (eight), *CG14118* (12), *brinker* (nine), *Scylla* (seven), and *sickle* (10 sites). Moreover, all the genes encoding endonucleases had more than one GAGA-binding site. Two other features of the probable GAGA target genes from the GO "apoptotic process" list should be highlighted: (1) more than half of them (35 of the 63 genes containing type 1 GAGA-binding sites and 30 of the 55 genes containing type 2 GAGA-binding sites) repress apoptosis; (2) at least one-third of the GAGA target genes in this dataset are developmental genes (30.16% of type 1 and 36.36% of type 2 GAGA target genes), for example, *abdA* (three GAGA-binding sites), *abdB* (four sites), *en* (two sites), and *sc* (14 sites). Most likely, it is these developmental genes that contribute to the observed higher density of GAGA-binding sites in the −500…+1 region of death-related lists of genes in the Flybase GO database.

In the $Trl^{R85}/Trl^{362}$ mutant testes, only 10 probable GAGA target genes from the Flybase GO list for the "apoptotic process" term were differentially expressed: endonuclease *CG14118* was 3-fold downregulated, whereas nine genes (*Orct, ft, lncRNA:Hsromega, Corp, fkh, Buffy, Abd-B, en*, and *pnt*) were upregulated. It is important to point out that it is impossible to obtain information about miRNA expression levels from our RNA-seq data; therefore, we cannot say anything about the expression levels of the 10 genes of apoptosis-suppressing miRNAs that contain probable GAGA-binding sites. miRNAs bantam, miR-14, and miR-278 and the miR-2 family, which includes miR-2, -6, -11, -13, and -308, are known to independently suppress apoptosis at the post-transcriptional level (*Jovanovic & Hengartner, 2006*).  In addition, among the target mRNAs of the listed

miRNAs, there are transcripts encoding regulators of other types of cell death, such as Fhos, a component of programmed autophagic death, and NijA, a contributor to necrosis. Downregulation of these miRNAs can activate translation of hundreds of their target mRNAs and launch cell death.

Similarly, of the 333 genes in the "programmed cell death" GO list, less than half of the genes (41.14%, *i.e.,* 137 genes, including 11 miRNA genes) contain potential GAGA-binding sites. Seventy-six genes (22.82%) contain GAGA type 1 sites, and 69 (20.72%) genes contain type 2 sites. The highest number of GAGA-binding sites is found in the −500…+1 region of genes *Drep3* (14 sites), *scute* (13), *CG14118* (12), and *sickle* (10 sites). Furthermore, as in the case of the "apoptotic process" term, the set of probable GAGA target genes contains developmental genes.

The GO list "autophagy genes" includes 200 genes, of which only 59 (29.5%) carry potential GAGA-binding sites. Forty-two genes contain GAGA-binding sites of type 1, and 32 of type 2. The largest number of the sites was detected in genes *AMPKalpha* (eight sites), *daw* (seven), and *hid* (six).

Thus, our results on the density distribution of GAGA-binding sites in 500 bp upstream regions of genes indicate that it is unlikely that GAGA as a transcription factor directly controls regulated germline cell death in *Trl*-mutant testes. Although we observed slightly higher density of GAGA-binding sites in the Flybase GO lists of genes corresponding to "programmed cell death", "apoptotic process", and "cell death" (Fig. 4), this result may be explained by the presence of developmental genes in these lists.

## DISCUSSION

The GAGA protein is a global regulator of the expression of thousands of genes and is involved in the modulation of transcription at several levels: chromatin remodeling, transcription, and RNA Polymerase II (Pol II) pausing (*Van Steensel, Delrow & Bussemaker, 2003*; *Tsai et al., 2016*). In our earlier work, we showed that a GAGA deficit in *Drosophila* testes leads to autophagic death of germline cells and testis diminution (*Dorogova et al., 2014*; *Dorogova et al., 2021*). In the present report, we addressed the participation of GAGA in transcriptional regulation of male germline cell death. We conducted an RNA-seq analysis of control and mutant testes and compared the corresponding gene sets to each other. Our analysis revealed some characteristics of the regulation of cell death in the *Drosophila* testes and, in particular, answered the question what role GAGA plays in the control of cell death in *Drosophila* testes.

### DEGs in *Trl*-mutant testes

An unexpected funding in the analysis of the transcriptomes is the predominance of upregulated genes over downregulated ones among the DEGs. As many as 69.18% of the DEGs proved to be overexpressed in the *Trl*-mutant testes whereas only 30.82% were underexpressed. It is known that transcription factor GAGA is a positive regulator of global gene expression (*Tsai et al., 2016*); consequently, we expected that its depletion in the testes should raise the expression of most genes. Indeed, in the larval imaginal discs carrying a similar combination of the *Trl*-null allele with a hypomorphic *Trl* mutation, 82% of
genes turned out to be underexpressed, confirming a positive role of GAGA in global gene regulation (*Tsai et al., 2016*). It is possible that the expression skew toward gene activation that we registered in *Trl*-mutant testes is not due to the GAGA activity as a transcription factor but rather to other aspects of its function. For example, GAGA is involved in chromatin remodeling and can modulate gene expression by changing nucleosome density in a promoter region (*Judd, Duarte & Lis, 2021*; *Fuda et al., 2015*). This notion is supported by our finding that the set of upregulated DEGs, according to FlyEnrichr, is enriched with the target genes of transcription factors Pc and Su(Hw) (Fig. S6). In *Trl*- mutant testes, more than half of the target genes of Trl, Pc, and Su(Hw) show upregulation (Fig. S6). Pc and Su(Hw) modulate gene expression at the chromatin level. Direct protein–protein interactions of GAGA with Pc as well as with some other components of the Pc complex are reported to enhance the activity of the Pc complex (*Poux, Melfi & Pirrotta, 2001*). No direct binding of GAGA with Su(Hw) has been reported but they can interact *via* Mod(mdg4) (*Melnikova et al., 2004*; *Ghosh, Gerasimova & Corces, 2001*). For instance, the interaction between GAGA and Mod(mdg4) is a possible mechanism governing gypsy insulator activity (*Melnikova et al., 2004*). Normally, the Pc complex and the Su(Hw) insulator repress large groups of genes at the beginning of spermatogenesis, thereby preventing premature differentiation of germline cells into spermatozoa (*Zhang et al., 2017*; *Feng, Shi & Chen, 2017*; *Glenn & Geyer, 2019*). GAGA may directly or indirectly take part in these processes; therefore, its depletion drives alterations in the composition of repressor and/or insulator complexes and induction of some repressed genes.

Another reason for the prevalence of gene activation in *Trl*-mutant testes may be the cancellation of Pol II polymerase pausing. Paused RNA polymerase II is located ~30–50 bp downstream of the TSS of genes associated with developmental control, cell proliferation, and intercellular signaling (*Tsai et al., 2016*). GAGA is enriched on promoters with paused Pol II (*Tsai et al., 2016*; *Fuda et al., 2015*). The groups of genes that must be rapidly and synchronously expressed in response to developmental or stress signals, such as heat shock protein genes, are normally inactive (*Vihervaara, Duarte & Lis, 2018*). For example, the *Hsp70* gene encoding a chaperone is highly repressed, but 1 min of heating is sufficient to activate it (*Tsai et al., 2016*; *Duarte et al., 2016*). Some researchers demonstrated a decline of the amount of paused Pol II as well as productive *Hsp70* transcription in the absence of GAGA (*Tsai et al., 2016*). Thus, the increased expression of approximately $^2/_3$ of genes in the *Trl*-mutant testes may be due to the downregulation/inactivation of GAGA as a chromatin remodeler or RNA polymerase pausing enforcer.

## Testis-specific genes

Normally, the GAGA protein is localized to the apical end of the testes in germline cell nuclei at early premeiotic stages (*Dorogova et al., 2014*). This period of spermatogenesis is characterized by a high level of gene transcription and synthetic activity. The volume of spermatocytes increases 25-fold, which requires considerable consumption of energy and resources (*Fuller, 1993*). Germline cells at premeiotic stages have transcriptome composition that is dominated by genes responsible for mitotic division and preparation for differentiation. In this regard, our data match the results of *Vedelek et al. (2018)*

who divided the testis into three parts—apical, middle, and basal—and analyzed the transcriptomes of each part (*Vedelek et al., 2018*). According to their data, the apical part of the testis is enriched with genes that are maximally expressed in other tissues, not testes. Testis-specific genes were detected mainly in the basal part and were found to play a part in sperm differentiation and morphogenesis (*Vedelek et al., 2018*).

In GAGA-deficient testes, germline cells are present only in the early stages of spermatogenesis; starting from the stage of spermatocytes, they begin to degrade and get eliminated (*Dorogova et al., 2021*). This means that the testes of *Trl* mutants actually consist of the apical part as a consequence of the early death of the germline cells. As we expected in this case, testis-specific genes are under-represented in the RNA-seq data on the *Trl* mutant (Fig. 1), in agreement with the data from *Vedelek et al. (2018)*. Among 2,602 testis-enriched genes from Vedelek et al.'s dataset, 148 genes were not detectable in our experiment, whereas the other 1,855 showed underexpression. Thus, the results of RNA-seq analysis of *Trl*-mutant testes confirmed the observed mutant phenotype, *i.e.,* the death of spermatocytes is induced before their differentiation.

A characteristic feature of our RNA-seq dataset is its enrichment with testis-under-represented genes (Fig. 1). FlyEnrichr GO analysis indicated that in the mutant testes, the upregulated genes were mainly metabolic genes (Fig. 2). We believe that this evidence may reflect increased cellular stress: when GAGA is in short supply, an imbalance develops in the expression levels of genes from different processes, for example, in the mitochondrial respiratory chain. The unbalanced amounts of products of such genes, on the one hand, drive additional induction of genes coding for the missing components. On the other hand, an excessive amount of proteins may not be recycled in time by the proteolytic system and can create insoluble protein aggregates that trigger the unfolded protein response, which alters the expression of many genes involved in endoplasmic-reticulum quality control. At the same time, mitochondrial dysfunction leads to a lack of energy in the cell and again upregulates metabolic genes. By contrast, in the absence of the GAGA protein, transcriptional regulation is disturbed, promoting cellular stress, activation of autophagy (in an attempt to utilize some proteins and remove destroyed organelles), and in the end, cell death. This notion is supported by the observed dramatic overexpression of stress-inducible genes and lncRNA genes: *e.g.*, the *Hsp70Ba* gene which encodes a chaperone participating in the response to heat shock and hypoxia and stress response-related lncRNA genes *CR34262*, *CR32010* and *CR45346*. Taken together, these data suggest that the observed cell death during spermatogenesis is not related to cell differentiation or seminal functions but rather is a consequence of metabolic aberrations.

## lncRNAs and miRNAs

Approximately 1/3 of our DEG set consists of lncRNAs, a class of noncoding RNAs longer than 200 nucleotides. More and more evidence has been accumulating about the functions of lncRNAs in numerous biological processes and in diseases (*Choudhary et al., 2021*; *Xu et al., 2017*; *Li et al., 2019*; *Deniz & Erman, 2017*). LncRNAs are predominantly localized to the nucleus and implement gene expression regulation at epigenetic, transcriptional, and post-transcriptional levels (reviewed by *Li et al. (2019)*). Underexpression of some lncRNAs

may result in abnormal embryogenesis, loss or decline of *Drosophila* fertility (*Wen et al., 2016*; *Li et al., 2019*). *Wen et al. (2016)* revealed 33 testis-specific lncRNAs involved in the regulation of spermatogenesis, four of them with potential GAGA-binding sites altered their expression in *Trl*-mutant testes. Their knockdown certainly contributed to the spermatogenesis abnormalities we observed, but it should be noted that these lncRNAs act at later (postmeiotic) stages of spermatogenesis as compared to the death of *Trl*-mutant spermatocytes. The knockdowns of CR43414 and TS15 *via* RNA interference impaired the polarization of spermatids and caused a lag or poor alignment of individualization complexes; the CR43753 knockdown resulted in an abnormal early phase of spermatid elongation; and the CR43862 knockdown yielded scattered and curled sperm nuclei in late spermatogenesis. By contrast, in the case of hypomorphic *Trl* testes, death occurred at the spermatocyte stage, and most cells did not reach meiosis

Recent studies have revealed important functions of lncRNAs in the modulation of autophagy, the regulation of metabolism and the stress resistance in *Drosophila* (*Li et al., 2019*; *Lakhotia et al., 2012*). Moreover, any dysregulation of lncRNA expression is enough to reduce stress tolerance. For example, lncRNA hsr $\omega$ overexpression as well as nullisomy or its RNA interference are lethal for most *Drosophila* embryos and first- or third-instar larvae under heat stress (*Lakhotia et al., 2012*). lncRNA hsr $\omega$ contributes to omega speckle formation (which is a spatial repository of key regulatory factors bound to their pre-stress nuclear targets in cells recovering from stress) and to regulation of the protein metabolic process (*Lakhotia et al., 2012*; *Lo Piccolo, Mochizuki & Nagai, 2019*). In *Trl*-mutant testes, we detected 3.35-fold upregulation of lncRNA hsr $\omega$ (Table S3) and huge overexpression of other stress response lncRNAs, *e.g.*, CR34262, CR32010, and CR45346. We suppose they may contribute to the aforementioned upregulation of metabolic genes. On the other hand, expression dysregulation of some lncRNAs may induce the autophagic cell death in mutants. It is known that the highest expression level of lncRNA lncov1 coincides with the autophagic cell death in the larval ovary of the worker bee *Apis mellifera* (*Choudhary et al., 2021*). In humans, overexpression of BRAF-activated lncRNA (BANCR) raises the LC3-II/LC3-I ratio, a marker of autophagy (*Xu et al., 2017*).

The regulation of cell death *via* the lncRNA–miRNA axis deserves special attention and requires further research. miRNAs are small noncoding RNAs 18–24 nucleotides long that control gene expression post-transcriptionally through mRNA degradation or translation inhibition (*Leaman et al., 2005*). Many miRNAs are implicated in the regulation of cell death: the largest miRNA family (miR-2, -6, -11, -13, and -308), miR-14, and bantam are inhibitors of apoptosis in *Drosophila* (*Leaman et al., 2005*; *Xu et al., 2003*); the mammalian miRNA-30a family attenuates Beclin-mediated autophagy stimulation (*Xu et al., 2017*). lncRNAs and miRNAs interact with each other at different levels: direct transcriptional regulation (some lncRNA sequences contain miRNA recognition elements [MREs]); lncRNA stability can be weakened by miRNAs; lncRNAs serve as miRNA decoys or sponges and can compete for target mRNAs; finally, lncRNAs can be a source of miRNAs (*Xu et al., 2017*). Thus, in *Trl*-mutant testes, the dysregulation of lncRNA expression may disrupt the balance of lncRNA–miRNA interactions contributing to germline cell death.

## Cell death genes

The question that we addressed in this paper is whether GAGA transcriptionally regulates cell death in *Drosophila* testes. GO analysis of the testis DEGs did not reveal any enrichment with gene groups associated with cell death. Moreover, the majority of cell death-related genes did not manifest changes in expression in the mutant testes (Fig. 3), and downregulated genes were prevalent among the differentially expressed cell death-related ones.

Because germline cell death in the mutant testes is autophagic, we previously suggested that *Trl* mutations negatively affect cell metabolism by preventing the necessary magnitude of macromolecule synthesis and cell growth (*Dorogova et al., 2021*). The lack of energy and macromolecules as a rule activates the TOR signaling cascade, which regulates autophagy. GAGA protein deficiency can affect the expression of either genes encoding components of the TOR pathway and/or factors regulating autophagy, which can also cause ectopic death (*Levine & Klionsky, 2004*; *Das, Shravage & Baehrecke, 2012*). Overexpression of *Atg2, Atg4, Atg5, Atg7, Atg8, Atg9, Atg16, Atg17,* and *Atg18* and inactivation of TOR kinase are necessary for autophagy activation by this signaling cascade (*Levine & Klionsky, 2004*; *Das, Shravage & Baehrecke, 2012*). We failed to detect transcriptional activation of TOR pathway genes in the present RNA-seq study, although some individual genes from this pathway proved to be down- and upregulated. For example, no *Atg* gene underwent a more than 2-fold expression change, but genes *stj* and *lft* involved in autophagosome maturation were upregulated 31.34- and 2.25-fold, respectively (Table S3). It is possible that TOR pathway activation in the mutants proceeds in a noncanonical manner and requires further study.

Next, we looked at whether potential GAGA-binding sites were present in the regulatory regions of cell death-related and stress-related genes (Fig. 4). We chose the region of 500 bp upstream of the TSS because it was shown previously that peaks of GAGA binding are situated within this region in the majority of GAGA target genes (*Tsai et al., 2016*). Moreover, stress-activated genes whose induction is dependent on GAGA have a strong tendency to contain GAGA-binding sites immediately upstream of the TSS, between positions $-100$ and $-50$ (*Duarte et al., 2016*). Our analysis confirmed the reports that the list of developmental genes is enriched with GAGA-binding sites (*Van Steensel, Delrow & Bussemaker, 2003*; *Omelina et al., 2011*). We detected no enrichment with GAGA-binding sites in the gene lists corresponding to terms "autophagic cell death", "autophagy", "lysosome", "mitochondrion", and "respiratory chain complex" in the Flybase GO database, but we did detect slight enrichment with GAGA-binding sites in GO gene lists corresponding to "programmed cell death", "apoptotic process", and "cell death". On detailed examination, it appeared that this increase in the density of GAGA-binding sites is attributable to the developmental genes present in the GO gene lists (Table S3). It should also be noted that approximately one-third of the genes carrying probable GAGA-binding sites from the "apoptotic process" gene list either have nonapoptotic functions in development or are implicated in programmed cell death during development or morphogenesis. For example, the main role of *scute* is the determination of sex and the development of the nervous system; *engrailed* is essential for posterior compartment identity and for compartment boundary formation and maintenance; and *Ecdysone receptor*

*(EcR)* launches both molting and metamorphosis (http://flybase.org/). Some of the GAGA targets belong to the group of genes regulating programmed cell death during development. For instance, the *Abd B* gene is involved in promotion of the apoptotic process associated with morphogenesis; *argos* is necessary for facet differentiation and programmed cell death during eye morphogenesis; *Scylla* acts as cell death activator during head development (http://flybase.org/); and *hid* activation causes the programmed apoptotic cell death during facet formation (*Hsu, Adams & O'Tousa, 2002*). Additionally, some genes have been added into the list of cell death-related genes according to a prediction of their function, which is not always accurate. For example, the *peanut* gene is classified as apoptotic because its mammalian homolog, *ARTS*, has a truncated isoform that is localized to mitochondria and actually contributes to the modulation of apoptosis (*Mandel-Gutfreund, Kosti & Larisch, 2011*). Our studies suggest that in *Drosophila*, the *peanut* gene does not have a truncated isoform, the PNUT protein is localized subcortically, and its overexpression or deficiency has no effect on cell death (*Akhmetova et al., 2017*; *Akhmetova et al., 2015*). Thus, we believe that the modest enrichment of official GO lists of cell death-related genes with GAGA-binding sites is a false positive result because it can be explained by the presence of developmental genes and genes with unproven contributions to the regulation of death in the gene lists. Only the "programmed cell death" list is truly enriched with GAGA-binding sites, not surprisingly, because it is known that the transcription factor GAGA helps to control many developmental processes in *Drosophila* (*Bhat et al., 1996*; *Dos-Santos et al., 2008*; *Omelina et al., 2011*; *Bayarmagnai et al., 2012*).

Taken together, these data indicate that it is unlikely that GAGA regulates cell death of germline cells in *Drosophila* testes at the transcriptional level. We think that the germline cell death observed in *Trl* mutants may be a consequence of imbalanced intracellular processes (primarily metabolism and mitochondrial functioning) that lead to cellular stress.

## CONCLUSIONS

Previously, we have shown that in *Trl* mutants, mass death of germline cells occurs during spermatogenesis. At the earliest stages, pathological changes are detectable in the mitochondrial apparatus of cells: hypertrophy and swelling of mitochondria, matrix decondensation, and crista degradation. *Trl*-mutant spermatocytes are subject to excessive autophagy and lysis at the premeiotic growth stage (*Dorogova et al., 2014*; *Dorogova et al., 2021*). Given that the *Trl* gene encodes the transcription factor GAGA, which governs the expression of many genes, we decided in the present study to test whether it regulates cell death at the transcriptional level. We performed RNA-seq analysis of testes carrying a combination of null allele $Trl^{R85}$ andhypomorphic mutation $Trl^{362}$. Examination of the results at the transcriptional level confirmed disturbances of mitochondrial structure and function and developing cellular stress in spermatocytes. Characteristic features of the RNA-seq dataset of *Trl*-mutant testes turned out to be (1) diminished expression of the testis-enriched genes that are essential for sperm morphogenesis at later stages, indicating slowed or abrogated spermatocyte differentiation; (2) greater expression of ubiquitous or

multitissue genes, among which metabolic genes dominated, indicating cellular stress; (3) opposite gene expression changes in many biological processes, such as the respiratory chain and programmed cell death. Nonetheless, we did not identify any specific signaling cascade whose activation could lead to the death of germline cells deficient in GAGA. Furthermore, we failed to detect enrichment with GAGA-binding sites within $-500...+1$ regions of genes corresponding to autophagy, cell death, or stress-related GO terms.

Altogether, these findings suggest that the mass degradation of *Trl*-mutant spermatocytes represents regulated cell death caused by the cellular stress that is probably a consequence of imbalanced intracellular processes, primarily metabolism and mitochondrial functioning.

## ACKNOWLEDGEMENTS

We thank ICIG SB RAS Genomics Core Facility and Gennady Vasilyev for RNA sequencing. The English language was corrected and certified by Shevchuk Editing.

### Funding
This study was supported by The RFBR Project No. 20-04-00496-a and The Ministry of Science and Higher Education of the Russian Federation Project No. FWNR-2022-0015. The funders had no role in study design, data collection and analysis, decision to publish, or preparation of the manuscript.

### Grant Disclosures
The following grant information was disclosed by the authors:
The RFBR: Project No. 20-04-00496-a.
The Ministry of Science and Higher Education of the Russian Federation: Project No. FWNR-2022-0015.

### Competing Interests
The authors declare there are no competing interests.

### Author Contributions
- Svetlana Fedorova analyzed the data, prepared figures and/or tables, authored or reviewed drafts of the article, and approved the final draft.
- Natalya V. Dorogova conceived and designed the experiments, performed the experiments, authored or reviewed drafts of the article, and approved the final draft.
- Dmitriy A. Karagodin conceived and designed the experiments, performed the experiments, analyzed the data, prepared figures and/or tables, and approved the final draft.
- Dmitry Yu Oshchepkov analyzed the data, prepared figures and/or tables, and approved the final draft.
- Ilya I. Brusentsov performed the experiments, analyzed the data, prepared figures and/or tables, and approved the final draft.

- Natalya V. Klimova performed the experiments, prepared figures and/or tables, and approved the final draft.
- Elina M. Baricheva conceived and designed the experiments, authored or reviewed drafts of the article, and approved the final draft.

## Data Availability

The RNA-seq data are available at NCBI: PRJNA785453.

The *Oregon R* sets are available at NCBI: SAMN23563622, SAMN23563623, SAMN23563624.

The *TrlR85/Trl362* sets are available at NCBI: SAMN23563625, SAMN23563626, SAMN23563627.

## Supplemental Information

Supplemental information for this article can be found online at http://dx.doi.org/10.7717/peerj.14063#supplemental-information.

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
