# Peer review of "The complex role of transcription factor GAGA in germline death during Drosophila spermatogenesis: transcriptomic and bioinformatic analyses"

_PeerJ, doi:10.7717/peerj.14063_

## Round 0.1 · original submission · Minor Revisions

Please heed both reviewers' comments in resubmitting a revised version of the manuscript.

Reviewer 1 ·

Basic reporting

In this work, Fedorova et al., analyse the transcriptome of the adult testis of control male fruit flies vs Trl hypomorphs in heteroallelic combinations using RNAseq. The authors do careful bioinformatic analyses of the data obtained from these experiments, and are able to attribute previously described Trl loss-of-function phenotypes to changes in expression of genes related to, for example, mitochondrial function and metabolism. The manuscript is clear, well organised, and it presents a concise hypothesis that is properly tested. Therefore, it would be adequate for publication after minor revisions, as listed below.

Experimental design

- The authors should describe how are testes dissected and processed for RNA extraction.
- In lines 196-198, the total number of DEG and those up- and down-regulated genes do not match.
- In figure S3, significance for each combination should be clearly stated (as presented in Table S2).
- For the same set of experiments, the number of biological and technical replicates is not indicated, this should also be presented clearly.

Validity of the findings

- The authors state that "official GO lists of cell death–related genes with GAGA-binding sites is a false positive result because it can be explained by the presence of developmental genes and genes with unproven contributions to the regulation of death in the gene lists" (lines 619-622), however, they also point out that the gene hid is one of the genes with the highest number of GAGA-binding sites on their list (line 426). Given that hid is one of the best characterized pro-apoptotic genes in Drosophila, these two statements seem to contradict each other.
- In this same regard, it is not clear from the text whether hid expression is altered upon loss of Trl. This is important because hid could be involved in the cell death phenotype observed in the testis. The authors also raise the interesting possibility that Trl may negatively regulate the expression of certain target genes. Given that hid has several GAGA binding sites, it would be an interesting scenario that upon loss of Trl, hid would be over-expressed, which could explain the phenotypes previously observed.
- The authors identify several ncRNAs as DEGs, which had been previously described as testes-specific transcripts and that furthermore have testes phenotypes when knocked-out. It would be interesting if the authors could explicitly mention whether they find GAGA binding sites in the proximity of these lncRNAs, and if not, if they could extend their analyses beyond -500bp only for these genes. Misexpression of these lncRNAs could be contributing to the Trl mutant phenotype, possibly directly downstream of Trl.

Reviewer 2 ·

Basic reporting

GAGA is an important transcription factor known in Drosophila as Trl. It has been previously published that Trl mutants experience massive degradation of germline cells in the testes due to autophagy death of spermatocytes. In an attempt to understand the phenotype of Trl mutant animals in the male germline, Fedorova et al. performed RNAseq of Trl mutant testis. Unexpectedly, most of the genes found in the RNAseq analysis are up regulated suggesting that the GAGA protein might play another non-canonical role or roles besides its well-known function on transcription. Unexpectedly, authors did not find a big representation of cell death genes in their RNAseq analysis that could explain the phenotype observed in Trl mutant animals. The RNAseq analysis revealed unexpected roles for GAGA protein for example during metabolism, and in long non-coding RNA regulation.


Minor correction
1. Check spelling of Gorilla in lines 201 and 205.

Experimental design

This an original primary research within the aims and scope of the Journal. The research is meaningful.

Validity of the findings

The manuscript is well written however there is a lot of speculation in the discussion. It would be better to tone down the interpretation of the data. Since the RNAseq analysis did not reveal a direct role of GAGA protein in cell death, it would be better to not justify the RNAseq analysis based on this particular phenotype.

Additional comments

I do not have additional comments.

---

## Round 0.2 · Minor Revisions

Dear authors: Please revise the manuscript as suggested by reviewer two, so that the manuscript can be deemed acceptable for publication afterwards.

Reviewer 1 ·

Basic reporting

All points stated in the previous round of revisions have been addressed by the authors.

Experimental design

No coments.

Validity of the findings

No comments.

Reviewer 2 ·

Basic reporting

Authors have responded satisfactory to my concerns. However the manuscript still have to be edited carefully. There are still some mistakes.

Minor corrections:

1. Drosophila sometimes is not in italics. For example First paragraph Results section and second paragraph in "LncRNAs and miRNAs" section.

2. Please eliminate: "now added into"in the following paragraph:

"27.7% of the DEGs in our transcriptome dataset were lncRNAs, we analyzed the lncRNA genes for potential GAGA-binding sites; the results are shown now added into Table S4 and Figure 4.".

3. Some paragraphs are very long. For example first paragraph "LncRNAs and miRNAs" section.

Experimental design

No further comments

Validity of the findings

No further comments.

Additional comments

No additional comments.

---

## Round 0.3 · Minor Revisions

Please indicate in the manuscript where the RNA-seq data produced in this paper is deposited, and available for public consultation, before the paper can be accepted.

---

## Round 0.4 · accepted · Accept

Dear Authors:

Your paper has been accepted at PeerJ! Congratulations!

Reviewer 2 ·

Basic reporting

Authors have responded satisfactorily to all the comments. The manuscript is ready for publication.

Experimental design

No further comments.

Validity of the findings

No further comments.

Additional comments

No further comments.